# Pedagogical Contexts When Gender and Emotions Intersect with the Body: Interviews with Feminist PE Teachers and Associated Professionals

**DOI:** 10.3390/ijerph192315510

**Published:** 2022-11-23

**Authors:** Irati Leon, Rakel Gamito, María Teresa Vizcarra, Ana Luisa López-Vélez

**Affiliations:** 1Department of Musical, Visual Arts and Physical Education, Universidad del País Vasco, 48940 Leioa, Spain; 2Department of Didactics, School Organization, Universidad del País Vasco, 48940 Leioa, Spain

**Keywords:** emotional well-being, diversity, emotional education, physical education, vocational training, gender perspective

## Abstract

The importance of approaching school physical education from the perspective of body and identity diversity is currently receiving increased recognition generally, and this is perhaps especially true in the Basque Country, Spain. A number of involved professionals are committed to emotional education from a gender perspective with the objective of facilitating emotional awareness through the body. The aim of this research project was to compile the experiences and reflections of relevant professionals in the field of education and, on the basis of this testimony, identify limitations with respect to current engagement in this area. We conducted fourteen in-depth interviews with key informants, and their transcripts were analysed using Nvivo software release 1.4, contrasting the coding two by two. The results provide a snapshot of the current context and highlight some of the existing discourses and challenges in the education system around this topic. We conclude that physical activity is a key area in which to work on emotional education and reinforce a gender perspective.

## 1. Introduction

There is currently a robust consensus that learning is a complex process that cannot be reduced to cognitive factors. The context present in many schools following the COVID-19 pandemic has brought the importance of multisensory education based on sound emotional education to the forefront, as well as the importance of interaction during the teaching-learning process [1].

The concepts of emotional awareness and emotional intelligence appeared as early as the 19th century and were included by Gardner [2] in his theory of multiple intelligences published in 1983. These concepts address a person’s cognitive capacity to be aware of their own emotions and those of other people and, thus, to reason and manage behavior in order to respond effectively to everyday demands [2,3,4].

Identifying, using, understanding and managing emotions is a key factor in personal development and well-being, and these capacities, in turn, are central to developing the ability to make sense of life dynamics, including social relationships [5,6,7,8]. Restricted emotional development can lead to a conflictive experience of gender, violent attitudes and non-acceptance of one’s own identity [9,10,11,12].

Socioemotional education can therefore be considered an effective practice to reduce social exclusion and improve coexistence [13,14] and is also a factor in student motivation [15]. Teaching emotional skills helps make students more responsible by improving their expression and externality. It assists them in developing more social strategies to solve interpersonal problems and conflict, understand others, make personal decisions, control emotions, develop communication and listening skills, and increase empathy. These competencies should be addressed in physical education and school sport [16].

Schools must offer safe spaces where the educational approaches developed include empathetic listening and provide the opportunity to work on emotions through the body and its acceptance [17]. Emotional intelligence is a multidimensional and dynamic cognitive capacity that can be reinforced and expanded through work on the body [16]. Working on emotions through movement and the body improves body awareness [18] and generates expressive capacity that improves communication [19]. This work can be carried out in spaces including school PE activities.

School physical activity is an assemblage including epistemological (usually science-based), material (human beings, sports equipment, etc.) and subjective (gender, ethnicity, etc.) bodies [20].

In this context, there is a general belief that gender discrimination is not a problem [21], despite the fact that participation in sports is generally segregated by sex. The underlying reality is that this division is an artefact of power imposed and maintained in a rigid, violent and hierarchical way so that people adapt to established social norms [22,23]. This sexual division is thus established as the common sense of heteronormative discourse and marks boundaries outside of which lie exclusion and discourses of otherness [24].

As for schools, here, inequality has been normalised, while at the same equality is narrated as an objective already achieved [25]. Gender stereotypes continue to pervade the organisation of schools across politics, academic content and even sports [26]. This is why it is essential that schools exercise a real commitment to establish measures that raise awareness and intervene with respect to exclusion and gender violence in the educational sphere [27,28].

To this end, Miralles-Cardona et al. [29] argue that a gender perspective and emotional intelligence must form the backbone of the whole school system across organisation, classroom methodologies, play, the organization of playgrounds, teachers’ attitudes and family situations. Sensitivity to gender is also essential in the field of communication and in relation to the publication of textbooks. The use of sexist language, educational materials, the androcentric curriculum and the lack of real and non-stereotypical references to elite female athletes all demand review [23,25,30].

In short, schools’ physical activity programs should offer students an opportunity to acquire the competencies needed to develop a critical consciousness of gender by fostering both corporal and emotional awareness [30,31]. It is important to use the language of emotions in physical activities and sports, as well as in daily educational practice [32], but this must be done from a gender perspective [25] so that education is transversal [33].

Feminist pedagogies can make an essential contribution to this transformation by contributing a gender perspective. This includes a critical and deconstructive intersectional approach to education that integrates a diversity of theoretical and methodological perspectives that identify and critically engage with interwoven power relations [34]. This includes accepting the imperative to decolonize and depatriarchalize discourses present in early childhood education [23,35,36].

It is also worth mentioning that physical activity facilitates the use of more cooperative and group-based methodologies, which help with both individual and collective self-perception and thus generate higher-quality relationships [37,38]. These qualities, in turn, share the values of feminism and make it possible to create inclusive, queer physical education that focuses on pleasure, affective experiences and the creation of new materialities. In essence, the objective is to go beyond imparting specific skills and to create space for affirming subjective experiences and developing activities and experiences adapted to all students, not only the privileged [20].

The critical engagement described above resituates physical education in a different framework in which it becomes necessary to understand the complexity of the exclusion generated by systems of power that operate in schools [39]. In this way, the diversity present in classrooms can be addressed from a deeply thought-out gender perspective. A real transformation of education based on experience and evidence is needed in order to practice feminist pedagogies in school physical education programs so that students can work on emotional awareness through the body and the acceptance of their own bodies. This demands a gender perspective applied to the body politic of physical education [24].

The sex/gender system is a structural axis of society and social relationships. Western thought deploys these binary categories of sex/gender to analyze reality by establishing two exclusive and dichotomous genders, which are complementary and hierarchical [40]. They are based on a theory created by science in response to political and social interests [41]. Gender remains a primary characteristic of identity [26]. While this is an important factor, identity is also impacted by intersections with other factors, including race, class, culture, geographic location, sexual orientation/identity, ability and body [42]. 

School, school physical education and school sport are, along with the family and the media, amongst the most influential socializing elements by which existing social systems are reproduced from a gender perspective [20,43]. Institutions ensure that social norms persist over time by means of various forms of oppression towards women and/or non-hegemonic bodies, thus fostering inequalities [44,45].

Along the same lines, heterosexuality and conformity with roles and gender expressions assigned to each binary identity seem to be the acceptable and desirable norms in socialisation processes [24,46,47,48]. This affects the emotional state of students who do not meet the cultural expectations with respect to gender, which is assigned at birth, and who are, therefore, perceived to be outside the norm [49,50]. Moreover, these students are constantly exposed to prejudice, discrimination, violence and bullying, especially in educational and/or sporting settings [49,51,52,53]. In this respect, the work of a number of authors [54] is of great importance, as it demonstrates the necessity of going beyond single approaches and focusing on wider educational areas.

The overarching aim of this research was to identify any possible links within the education system between the two central focus areas. How is education in emotional intelligence approached from a gender perspective? How is it to engage with gender from the emotional body? How do these two dimensions intersect in school physical activity programs? What impact can different more or less appropriate approaches have on students’ academic and life processes?

This article highlights the need to curb inequality in educational and sporting environments by taking into account the voices of professionals who work in education from a feminist perspective. Through their experiences and reflections on engaging with the question of emotional intelligence and working from a gender perspective, we describe several different initiatives and proposals that address current shortcomings in schools and school physical activity programs. This provides an overview of the dominant discourses around gender perspective. As such, the specific objectives for this paper are: (1)To identify and understand the intersections present around the question of emotional awareness as approached from a gender perspective in the discourses of professionals in the field of education and school PE programs;(2)To identify the factors that influence the current school context in terms of emotional awareness and exclusion linked to gender diversity, as well as the tendencies observed in educational practices in school physical education;(3)To highlight and elaborate on the challenges facing schools seeking to address the lack of a gender perspective and to commit themselves to equality in a real, transversal and intrinsic way.

## 2. Materials and Methods

### 2.1. Design

This paper presents a comprehensive case study describing some work centred on emotions being done from a gender perspective in schools in the Basque Country (Spain). We considered a case study to be the optimal method because these seek to understand what has happened in a given development and context and focus on causes and unique characteristics [55].

Through a case study, events can be analysed from different points of view in order to understand and reveal the complexity of an issue [56]. Ultimately, the case study we present in this paper provides value in terms of understanding the dynamics of change while also describing and documenting the current context [57].

The information collected was qualitative in nature because we used an explanatory method that offers “a more global and comprehensive approach to reality” [55] (p. 76). The aim was to achieve a holistic approach using a more flexible design [57]. Immersion in context and the interviews allowed us to identify meanings freely associated by participants with particular signifiers [58,59].

### 2.2. Participants

In-depth interviews were conducted with fourteen professionals in the education sector who work on the integration of gender perspective in school contexts from preschool to secondary school (Table 1).

The interviewees included: Eight activists from feminist social movements;Eight school teachers: three involved in early childhood education (0–6 years), three in primary education (6–12 years), and two in secondary education (12–16 years). Six worked in the field of physical education;Two people who are part of the feminist secretariats of two different trade unions;Four people who were involved in the implementation of coeducation plans as authors, supervisors or trainers. Two were involved in the “Skolae” Coeducation Plan developed by the Department of Education of Navarra, and two participated in the Equality and Coeducation Plans (2013–2016 and 2019–2023) drafted by the Basque government;Four teachers undertaking placements, vocational training and inter-university exchange.

All interviews were coded “I,” and each participant was assigned a number chronologically, according to the date of each interview (see Table 1). Thus, each interview can be identified by a unique alphanumeric code. The codes and further details about the respondents are presented in Table 1.

### 2.3. Instruments

We used in-depth interviews for data collection. To this end, we prepared a script with open-ended questions in order to facilitate a fluid dialogue and unrestricted expression [57,59]. Interviews were recorded and then transcribed for analysis.

Participatory dialogue, a qualitative methodology based on questions and answers, is an important strategy for understanding the world and identifying concrete knowledge [60]. This space of dialogue seeks to generate narratives that provide quality information centred on ideas, reflections and experiences [58,61] through sharing intersubjective experiences among equals [62,63].

The themes and questions included in the interview script were based on three distinct axes related to the research objectives: (1)A description of interviewees’ everyday work practices and their reasons for working on emotional awareness and from a gender perspective;(2)Their reading of the current context and current levels of awareness with respect to gender and emotional awareness through the body at schools in general and in school physical activity programs specifically;(3)Lines of work (experiences) and goals/targets (proposals) aimed at addressing gender and emotional awareness and the relationship between these and feminist pedagogies.

We also used field notes to collect additional information that emerged during the interviews [63]. The observations recorded in field notes were useful when it came to recalling the main ideas, listing them and establishing the structure of categorisation.

### 2.4. Procedures

Initial contact with participants was made via email, a key communication tool during a period in which the Basque Country was severely impacted by the global COVID-19 pandemic. Once the date, time and medium for interviews were agreed on, we sent participants an information pack detailing the ethical principles of the research and the procedures for the confidential treatment of the data. Interviewees subsequently granted written informed consent to take part in the research. All the interviews took place in private spaces that were comfortable for the interviewees. In most cases, this was an office in the school, university or other institution where they worked.

Initially, we carried out an intentional selection of interviewees [64], as we had identified a number of experts who could act as high-quality informants in advance. In the second phase, we expanded the sample using the snowball technique [65,66], as interviewees participating in the first round of interviews were able to refer us to other experts and people of interest in areas related to the research topic through their activist and professional networks.

### 2.5. Analysis

The interview transcripts were coded on the basis of a system of categorization (Table 2) with the help of the analysis software QRS NVivo release 1.4. This software offers several advanced options for qualitative data analysis and is one of the most widely used worldwide. We chose to use this tool because it provides “rich data management, with ample ability to edit, visualize code and link documents as they are created, coded, filtered, managed and recorded” [67] (pp. 7–8). Details of the analysis of qualitative information that we carried out are depicted in Figure 1.

We used the system of categorization in Table 2 to analyse the semantic content of the interview data [59,69] through a systematic inductive-deductive process; deductive because it was based on the literature review, which guided the research design, and inductive because it took into account the new ideas that emerged in the dialogue with interviewees. The ideas isolated in each paragraph of the interview transcription were organised into dimensions, categories and subcategories [70].

We established the dimensions and categories as follows: Emerging discourses: discourses linked to a gender perspective and emotional awareness from corporal experience. These discourses had a greater presence among professionals in the field of education:◌Reproduction of the existing system.◌Transformation.
Current context: the factors currently conditioning the impact of gender perspective and the development of emotional awareness in physical education programmes and pedagogical practices in general.◌Commitment.◌External decisions.◌Communication.
Challenges: the structural challenges facing schools and school physical education programmes in relation to generating transversal and intrinsic responses to emotional and corporal work from a gender perspective to achieve real equality:◌Analysis.◌Materials.◌Awareness.

We counted the number of references, considering each paragraph with the same meaning as a unit of measurement, and these units of measurement were compiled into categories thanks to the qualitative information processing programme Nvivo release 1.4, as the programme itself counts the references collected in each category. Table 2, in the column to the left of the number of references (no. ref.), lists the number of resources (no. rec.), which corresponds to the number of interviews that mention that category. Participants’ confidentiality was guaranteed at all times.

The principal strength of the results does not, however, reside in the percentage figures generated or the computer-based analysis. Instead, the process of data categorisation, which consists of identifying common themes, is key to obtaining results relevant to the research objectives [71]. Given this understanding, in a qualitative study, the number of references is just one piece of evidence. Results and analysis should not be determined solely by the number of responses but instead by the relevance of these responses to a given topic. Therefore, a category without a significant quantitative presence can still be important if it contains a pertinent statement [72].

### 2.6. Trustworthiness

Before the coding and categorisation process, several meetings were held between the researchers in order to establish consistency between coding systems, assessed by a method called inter-observer agreement [73]. Reliability reached 87%, a range considered optimal [74]. In addition, the Nvivo release 1.4 programme calculated the kappa index [75], which analyses the association between coders, by taking two by two and obtaining an average of K = 0.76, with a significance level of *p* < 0.001. For the subsequent analysis, the interviews were coded using double-blind crosschecks, as two different researchers coded each interview.

In order to ensure trustworthiness, we applied a process of triangulation or crystallisation, that is, a process of contrasting the voices, in order to obtain a more real and complete picture and a better understanding of the meanings attached to each construct [76].

A lens acting as a prism to refract and project light in different patterns, colours and directions is an appropriate metaphor to understand this process. Each interview was a unique lens that reflected different perspectives on the case being analysed [77]. This metaphor respects the principle that “we experience the same places, but we refract them through different professional eyes, genders, sensibilities, biographies, spiritual and emotional desires” [78] (p. 135).

## 3. Results

The dimensions established in the system of categorization responded to the objectives of the study and analyzed the discourses emerging in school (273 references); the current context of work on emotional awareness from a gender perspective in the classroom and in school physical exercise programs (90 references); and the pending educational challenges in relation to the subject studied (321 references). Table 2 presents the number of references and the percentages corresponding to each dimension and each category.

### 3.1. Emerging Discourses

Emerging discourses on school and school physical exercise programs appeared in 35% of the testimonies. Interviewees affirmed that schools reproduce the system but also provide perspectives with transformative potential.

A total of 12% of the interviews defined schools as spaces for the reproduction of established binary norms. References to false or illusory progress in the inclusion of a gender perspective and emotional awareness (4.2%) stand out as, according to a number of interviewees, schools are institutions that respond to market needs (2.3%) that do not leave much space for reflection and the acceptance of diversity. The system is designed so that students and teachers follow homogenising patterns and rhythms with little room for individuality.

*The pedagogies used respond to the needs of a capitalist productivist society. This becomes more obvious as schooling progresses, as there is more and more content and less and less values, less emotional awareness, less empathy, less importance given to personal identity and, on top of that, teachers’ salaries are higher and they are better valued. The system places more value on rationality and productivity and puts less emphasis on values and emotions addressed in earlier years, such as creativity. The education system is designed (for children) to sit down and work for 8 h and to see sport as a spectacle*.(I8_19/06/30)

Moreover, in the current era, political correctness prevails, and, as a result, a large number of professionals in the field of education define themselves as feminists. However, there is resistance to trans* discourses that are not hegemonic. This demonstrates, once again, the lack of progress at the educational level and the fact that school physical activity programmes still perpetuate the binary system of gender (5.5%).

*The structure has not changed at all, so, someone might sleep for 100 years and when they wake up they wouldn’t understand society, but they would understand schools and school PE because the structure is still a patriarchal structure*.(I13_19/11/30)

*The education system in general and physical education in particular are part of the system and reproduce the existing social order very effectively. They even dare to use approaches that seem progressive, but what actually changes? Maybe it changes the way children act as girls or boys, maybe it is a bit more flexible, but, in general, in very few places can they break with the heavily stereotyped norms in school and sports*.(I6_19/03/26)

By contrast, several interviewees claimed that schools operate as spaces for transformation (23%). All interviewees talked about feminist pedagogies (5.5%), including the importance of including different points of view (7.6%) and a critical gender perspective (9.9%) that helps generate empathy and improves coexistence between diverse identities on the basis of healthy bodily and emotional awareness without fear or insecurity.

*Our commitment as a union is not fall into the trap of the tolerance, to make the different and the unknown an example. If we question the norms, we have a better chance of being. This is what these pedagogies (feminist pedagogies) propose*.(I9_19/10/01)

However, as most cases involved political demands, they were not understood as pending tasks or challenges yet to be adequately addressed by the education system.

### 3.2. Current Context

In general, education professionals’ commitment (11.5%) to break with binarism and offer bodily experiences that facilitate emotional development from a gender perspective is minimal. There is a lot of talk but little interest in reviewing educational practices, including classic school PE activities, or in modifying the language which underpins ways of thinking (7.6%). Role model figures do not act collectively (0.9%) and usually, only particular individuals with specific training are involved in coeducational or intersectional projects (3%). Thus, the development of plans and proposals is restricted and eventually fails because there is no collective commitment.

*It is very frustrating to see that the person looking after the children is not able to play with them because their own clothing is not appropriate. They (carers) wear heels and mini-skirts and they can’t play with the children. In my personal work it is important to be comfortable and to set an example*.(I10_19/10/15)

*Some initiatives continue because there are people who are willing to work from a gender perspective and on emotions, but there is no structural support* …(I1_28/02/08)

External decisions (6.4%) were also identified as being important. We documented criticism focused at administrative and governmental level institutions (2%) as interviews stated that there was no real political will to work on these issues in the classroom, and there was no serious interest in providing teachers with resources. Institutions should ensure that specialised training is compulsory.

*I am convinced that it’s political cowardice, pure and simple. I have to say it, because that’s the reality. They use equality to paint over things a bit from time to time, for particular events. Every time they need our vote, women are brought up, equality is mentioned, but then in the end it’s not real, they don’t really care about it on the inside*.(I13_19/11_30)

Unions and other social agents (3.3%) also have feminist education on their to-do lists. Both unions and university-based actors (1.1%) have done a lot of work to create school materials in which gender diversity and non-normative bodies are present in order to promote students’ emotional well-being. Some interviewees testified that while education has a significant impact on the wider social context, it is not at the top of the political agenda of the feminist movement.

*[…] in my subject there is a relationship with gender. My students will graduate to be pre-school teachers and we have material to address and talk about gender, directly related to coeducation*.(I4_18/03/12)

*The key is education and we in the feminist movements have forgotten it’s not on our agendas, it’s not part of our demands. We are more focused on violence, sexualities, sovereignty… and now the wage gap and domestic workers. Education? We always say that the fundamental basis of a response to violence is education, but* …(I8_19/06/13)

With respect to communication (6.4%) about emotions and the body, participants related this directly to families and the difficulties involved in building bridges between generations and decentring the nuclear family model.

*They [families] are the pivotal nexus between parents and children. Therefore, the aim of the school should be to empower parents to help them take care of their relationships. If they talk at home and there are answers, they look for answers together*.(I14_20/03/02)

The basis of the problem lies primarily in the lack of awareness around gender diversity, acceptance of the body and its influence on emotions.

### 3.3. Challenges

Three challenges (41%) were identified: analysis (15.8%); material (12%); and awareness (13.2%).

In terms of analysis, the collective processes (5.3%) needed to establish feminist transformative projects in the classroom require spaces for systemic critical reflection and knowledge exchange (5.7%), as well as time, training and resources (4.8%). It is essential to work on the basis of critical thinking and subjectivities, to analyse privilege and oppression, and to engage in ongoing critical reflection on the work being done. If this is not the case, only the same few interested people will ever be involved.

*We need space for collective reflection… critical reflection from a gender perspective*.(I2_10/02/13)

*[…] there are many educators who do a lot of work, who want to do things differently and who put their bodies and their lives into it, but in the end individual initiatives come to nothing*.(I6_19/03/26)

*What educational institutions should do is to make training compulsory for all teachers, because if it is optional, it’s always the same people who go*.(I3_18/02/15)

The playground and how it is used was a recurring theme when addressing free play from a gender perspective. It was observed that certain inertias cannot be broken unless there is a will to do so. Many interviewees talked about transformation projects implemented in school playgrounds, but they also mentioned the need for work to be done consciously while thinking about the consequences from an interdisciplinary perspective.

*I find the inclusive playground very interesting from an architectural and spatial point of view. Normally in the playground there is a football pitch and sometimes a basketball court. I think it is very valuable to work on inclusion in all senses. In changing our perspective we can change, for example the playground, it’s a great tool which should be used in all schools, but this has to be done consciously, [while thinking about] where they are placed and why, where they [children] play and feel comfortable with the proposals we make*.(I12_19/11/26)

*Given the new plans drafted by the Basque Government, I hope that dynamics will be included in schools, because a coeducational observatory is needed to monitor things that are changing in schools*.(E1_28/02/08)

In terms of material (12%), interviewees recognised progress in recent years, but still observed the appearance of outdated stereotypes. In physical activity, sports generally remain segregated into stereotyped gendered categories, and very little work has been done on emotional awareness from gender identity or mental health perspectives (8.9%). Small discriminatory acts (3.1%) are difficult for pupils to identify, and the changing rooms are still the focus of contention.

*I arrived at the new school and the changing rooms were separated. How can that be? Our view is that to build healthy relationships we need to know our bodies: physical, emotional and cognitive and learn to relate to them, to relate to each other in a healthy way, that’s our goal and so, for example, we use the showers to see, respect and accept each other*.(I11_19/11/18)

*We have to go beyond the existing material and we have to create new material ourselves. In general, material is directed and stereotyped. We are winning this struggle little by little… We teachers are using other materials, but we continue to fall into the same stereotypes when in reality we want to get away from them*.(I10_19/10/15)

The above is related to the larger challenge of overcoming a lack of awareness (13.2%). Intersectionality (3.7%) is an important transformative element because it is needed to understand that people are diverse and that we cannot fall back into discourses of inclusion delivered from a condescending point of view (4.8%). It was also seen as necessary for professionals to carry out processes of personal reflection.


*Inclusion is not just taking into account different rates of advancement or rhythms for learning cognitive and motor skills, or facilitating class participation for people with functional diversity, or attending to the needs of those who have an illness, or offering support to students who are doing badly or those who are still learning the language used in the school, but also about taking into account all differences, understanding that we are all different. Intersectionality tells us that identity is complex, that it is not simple, that many things intersect. How can we include everybody? How can we do it? How do we think about it, how do we plan it, how do we manage it?*
(I5_18/03/14)

*With the plan, people are forced to reflect on their personal lives so it doesn’t all just stay in the classroom. Working on your awareness from your own experience is not the same as doing it from watching television. Training should be more oriented towards that path*.(I8_19/06/30)

In many cases, equality, diversity, bodies and emotions are not fully embedded discourses but instead loose ideas that come up only on specific occasions.

*The 25th arrives and everyone is in an uproar, it’s the day of the good deals, no kidding! What good deals? What we need are healthy relationships, with ourselves, so that we feel free and confident in ourselves. Constructing this is very difficult, because you need to be 100% present. Very often we are only in our heads and not in our bodies and we don’t live in the now*.(I11_19/11/18)

*We need to start work when they [students] are young. We want to incorporate a gender perspective and work on emotions in infant, primary and secondary education. Preschool education is the base and we can build the other stages on it*.(I7_19/06/06)

We need to change the way we look at things, stop talking about otherness, and work together with other perspectives that deal with different oppressions. A gender perspective (4.7%) must permeate different spaces from an early age with an interdisciplinary prism, starting with the analysis of our own lives.

## 4. Discussion 

The role of a gender perspective in the field of sports and education must be addressed urgently from a critical point of view, with emotional education and body awareness being central concerns. The current social context facilitates the development of strategies based on critical feminist awareness. Therefore, it is evident that this social context influences not only pedagogical proposals but also the way issues are understood, the acceptance of projects and the allocation of resources.

Schools and physical activity continue to be fundamental tools for the reproduction of the dominant system’s values. Competitiveness and gender binary norms are still hegemonic [22,23,24,25,26,28], and while politically correct discourses are present, they produce little transformation without any kind of emotional or bodily awareness.

We observed a unanimous consensus that feminism is fundamental and that working from an intersectional perspective, including transfeminism, de-colonialism, anti-racism, anti-ableism and queer theory, is fundamental [20,35]. Feminist pedagogies are beginning to permeate into classrooms, and this helps educational proposals centre emotions and the body.

These positions lead to the recognition that physical education faces a reality in which contexts and relationships of exclusion are complex [24,39,54]. Therefore, inclusion has to address other important issues and incorporate these struggles from an intersectional paradigm [34].

On the other hand, emotional education needs to be strengthened as a basis for working on issues of exclusion, oppression and conflict, as well as motivation [1,15]. Creating an empathetic school environment and opening opportunities to create alliances depends primarily on emotional education and the teaching of emotional skills, which can be developed through physical activity [13,14,16,29,37,38].

A cross-sectional perspective on the education system is indispensable, both in terms of implementing a gender perspective and emotional education [29]. This perspective must be implemented at all levels, including both content and role-model behaviour [52], as well as being present in all spaces across all age groups. In other words, a multidisciplinary, multisensory [1] and comprehensive approach is needed [22,27,33].

Systematisation is another urgent task because the implementation of a gender perspective continues to depend solely on activist teachers with feminist awareness instead of on collective processes or institutional school commitments [29,33]. Schools need to be provided with tools and time, as well as training, as these are key to raising awareness of gender, mental health and emotional education [25,29]. Beyond this, it is also important for schools to show that other models are both desirable and achievable in order to educate critically about the system that structures school organisation [25,30].

As to the university, this institution does tackle the issues brought up in this paper in various spaces and projects. However, there remains a need to systematise this in the syllabus delivered to future generations of teachers. The greatest challenge continues to be communication between different educational agents, establishing networks and working collectively on shared projects.

Families and communication with them also emerged as a subject of primary concern, as they are the most important agent in the schooling processes [43]. Thus, it is vital to provide families with information so that they can work on certain issues at home. Again, creating spaces for reflection between different agents is important, but the educational system makes it difficult to use these spaces in practice [25].

The need to work on emotional education from a gender perspective and on gender from the emotions forces us to take into account body and identity diversity and the impact that these have in terms of mental health and emotional well-being [49,50,51,52,53], placing importance on discourse critical towards gender norms. Similarly, emotional education centres on the body, experiential learning and well-being. This fills out otherwise empty discourses sold as progress with transformative educational practices [16,17,18,19].

## 5. Conclusions

The conclusions we obtained indicate that school physical activity programmes are key when it comes to working on emotional education and the implementation of a gender perspective in a way that emphasises the experiential, the corporal and well-being founded on mutual respect and care. Therefore, this educational area is a space where work can be done, collectively and in the community, to address complex contemporary realities from early childhood education through professional training.

Future research could broaden the scope of this paper and incorporate other perspectives, as well as the voices of students and families, thus expanding the sample of respondents quantitatively and qualitatively, as the number and profile of the interviewees were the limitations of this project.

## Figures and Tables

**Figure 1 ijerph-19-15510-f001:**
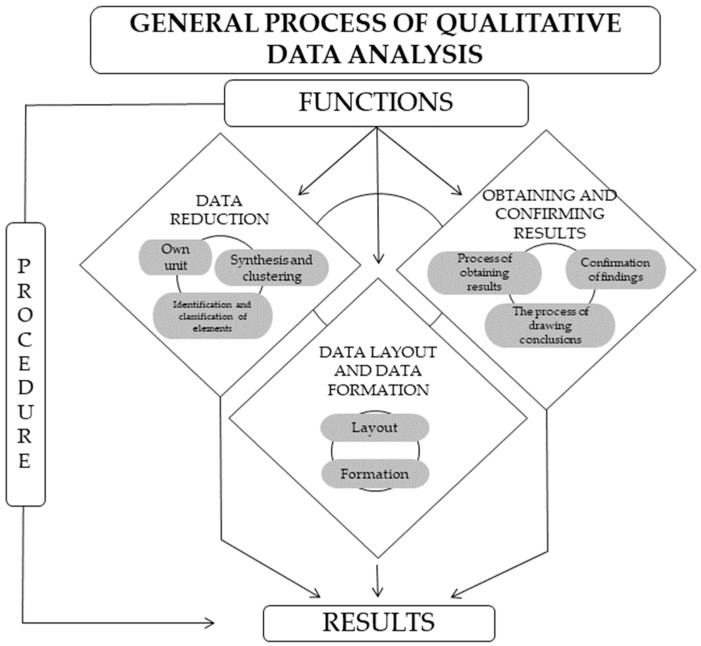
General process of qualitative data analysis. Adapted by the authors [68].

**Table 1 ijerph-19-15510-t001:** Participant details.

**1st Phase**	**I1:** Secondary physical education teacher. Head of a training and innovation support centre in the public education system in the Basque Autonomous Community. Author of the 1st Plan for Equality and Coeducation drafted by the Basque government.
	**I2:** Educator working in early childhood education (students 0–3). Member of the training team of the Consortium of Nursery Schools of the Basque government.
	**I3:** Activist in the feminist movement. Physical education teacher.
	**I4:** Lecturer and researcher at Mondragon Unibertsitatea (MU) with expertise in gender.
	**I5:** Activist in the feminist movement. Lecturer and researcher specialized in gender at the University of the Basque Country (UPV/EHU).
**2nd Phase**	**I6:** Former early childhood teacher (students 3–6) enrolled in vocational training. Member of “Bilgune Feminista” (feminist organization of the Basque Country).
	**I7:** Former teacher. Author and consultant for the Equality and Coeducation Plans I and II drafted by the Basque government for 2013–2016 and 2019–2023.
	**I8:** Activist in the feminist movement, Bilgune Feminista and the association Emagin (feminist research and documentation centre in the Basque Country) Trainer in the ongoing teacher training program established in the Skolae coeducation plan implemented by the Department of Education of Navarra.
	**I9:** Activist in the feminist movement. Primary school physical education teacher. Member of the feminist secretariat of the Euskal Herriko Steilas trade union.
	**I10:** Activist in the feminist movement and Bilgune Feminista. Preschool teacher (students aged 3–6).
	**I11:** Primary school physical education teacher.
	**I12:** Former feminist activist. Former participant in the feminist secretariat of the LAB trade union in the Basque Country. Primary school physical education teacher.
	**I13:** Feminist activist. Secondary school teacher. Author of the Equality and Coeducation Plans I and II of the Basque government for 2013–2016 and 2019–2023 and the Skolae Coeducation Plan implemented by the Department of Education of Navarra.
	**I14:** Member of the Arremanitz cooperative (cooperative for good treatment and parity) and former lecturer and researcher at the University of the Basque Country (UPV/EHU).

Source: Authors.

**Table 2 ijerph-19-15510-t002:** Categorical system and number of references.

Dimension	Category	Subcategory	Rec No.	Ref. No.	%
			14	273	35
Emerging discourses	Reproduction		14	92	12
		False progress	14	32	4.2
		Market needs	8	18	2.3
		Binary system	14	42	5.5
	Transformation		14	181	23
		Feminism	14	43	5.5
		Intersectionality	14	60	7.6
		The need to question	14	78	9.9
			14	188	274
Current context	Commitment		12	90	11.5
		Review and changes	6	60	7.6
		Project involvement	11	24	3
		Mentors/guides	3	6	0.9
	External decisions		11	50	6.4
		Administrative and governmental actors	6	16	2
		Trade unions and social partners	5	25	3.3
		The university	7	9	1.1
	Communication		12	48	6.1
Challenges			14	321	41
	Analysis		14	124	15.8
		Collective processes	14	42	5.3
		Analysis	13	45	5.7
		Resources	13	37	4.8
	Material		14	94	12
		Stereotypes	14	70	8.9
		Discrimination	14	34	3.1
	Awareness		14	103	13.2
		Diversity and inclusion	14	37	4.8
		Intersectionality	14	29	3.7
		Gender perspective	14	37	4.7
			**14**	**782**	**100**

Source: Authors.

## Data Availability

Not applicable.

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
