# Peer review of "Pedagogical Contexts When Gender and Emotions Intersect with the Body: Interviews with Feminist PE Teachers and Associated Professionals"

_ijerph, 2022, doi:10.3390/ijerph192315510_

Round 1
Reviewer 1 Report
Thank you very much for the opportunity of reviewing this manuscript. It was really interesting to read the soundness and connection of these topics. Since there is a lack of research being conducted in PA/PE, especially in schools.
I would recommend to add the work of Heather Sykes, Dillon Landi, Patricia Vertinksy in your intro and discussion, since their area of expertise is diversity and PA/PE.
Other than that, I recommend this manuscritp to be accepted in its form.
Author Response
Dear Reviewer,
Thank you very much for your feedback. On your advice I have incorporated the work of Heather Sykes and Dillon Landi in the intro and discussion. Thank you very much for these interesting references.
I attach the manuscript with the change control so that you can appreciate the change.
Sincerely yours,
Irati Leon

Reviewer 2 Report
Authors investigated the experiences and impressions of professionals in the educational field to find the existing shortcomings, and therefore to be able to work on emotions considering gender perspective by means of in-depth interviews. This work highlights the intimate relationship between emotional education and the gender perspective (i.e., the first is pivotal for emotional education and the gender perspective
This work could shed light on the psychosocial – and educational – students’ needs. Despite all, major revisions are needed to significantly improve the quality of the work.
Title: The title is not clearly informative since it not describes the direction of the evaluation that authors conducted. Title should also anticipate the results of the main outcomes to be “captivating” to readers’ eyes.
Abstract: Abstract conforms to informative conventions, but it is not clearly structured and it is not stand-alone.
Authors should provide general background, the rationale of the study, methods, the main research findings and a robust conclusion. For example, a good rationale is missing: this might improve readers’ comprehension toward the reasons that have driven the work. Why did authors conduct this study? What is the novelty aspect of this investigation? Then, methods are not clear, as well at all the steps of data analysis are missing. The results are clear at all, and their interpretation is absent. It is worth to note that the abstract should be a stand-alone object: the way it is presented is not sufficient to reach this status.
Introduction
Overall, the section appears clear and easy to follow. However, some parts need to be modified to improve its quality.
- The first part of the introduction is dedicated to the construct of emotional intelligence and emotional awareness, that assumes the principal role. In this line, suggest inserting a brief “digression” on the role of emotional intelligence on scholar performances: for example, the work of Cesari et al (2021) highlight the link between emotional and cognitive aspect of learning (i.e., social engagement, flow, and presence) during the mandatory form of online learning during pandemic (Cesari, V., Galgani, B., Gemignani, A., & Menicucci, D. (2021). Enhancing qualities of consciousness during online learning via multisensory interactions. Behavioral Sciences, 11(5), 57.),
- The introduction of gender perspective considering the important role of sport should be better explained. I suggest the following work, that also explain the role of gender stereotypes in many fields (including academic, politic, sport): Ellemers, N. (2018). Gender stereotypes. Annual review of psychology, 69, 275-298.
- Again, I have to highlight that a good rationale is missing in the introduction section. Authors should insert a valuable argument able to justify the conduction of the study.
I suggest inserting the rationale after the following sentences “Hence, the importance of becoming emotionally aware through the body and focusing the educational practice of physical education on those professional discourses that defend emotional education from a gender perspective is underlined.”
- In addition to the aims, I strongly recommend a brief paragraph describing Research Hypothesis, if any (i.e., was the study hypothesis-driven or data-driven?)
Research Methodology
- Authors should mention the criteria for participant enrollment, if any (i.e., were there any exclusion criteria for the enrollment? I suppose that non-mother tongue participants might constitute an exclusion factor). What about the settings in which participants complete the interviews? Were there any rules to avoid experimental biases (place and time of the day to complete the interviews, aloneness of participants, and so on)?
- When authors describe the instrument (i.e., the interviews), they should insert a valid rationale for the type of open questions submitted to participants. Moreover, authors should justify the choice of non-structured interviews instead of the structured ones
- The section “procedures” appear quite ambiguous, especially the steps regarding the enrollment of participants (are “intentionally” and “snowball technique” enrollment a reliable method for the interview?
- Authors reported the software used for qualitative data analysis and the main steps that led to the final categorization of dimensions, categories, and subcategories: I suggest a more accurate description of the procedures (the graphical description by means of flow diagram could help readers to better understand the procedures). Moreover, authors should justify the choice of NVivo release 1.4 software from the pool of qualitative data analysis tool.
Results:
Results section appear clear and easy to follow. I suggest improving the quality of the tables and to insert the sentences from interviews in a dedicated section in Supplementary Materials
Discussion
The section should be heavily implemented with a solid background. As it is described, it appears to be a mere paraphrase of the results. The discussion section should be an in-depth analysis of the current state of art that agree or do not agree with the results. Currently, I can define the discussion section as “shallow”.
Conclusions
The section is clear, but I recommend inserting a paragraph regarding the study limitations
Other comments
-I kindly invite authors to get the manuscript reviewed by experts in their field (Manuscript Editing Service): this will allow to significantly improve the linguistic aspect of the paper, thus facilitating the reading;
- I also recommend paying attention to the graphical aspect of the manuscript by providing high quality of tables
Author Response
Dear reviewer,
Thank you very much for your comments. On your advice, the following changes have been made to the manuscript:
- All sections have been modified, incorporating new content to create a better link (e.g. in the introduction and dictionary sections), as well as more specific information (especially in the method section) and correcting those that were not entirely appropriate.
The manuscript has been proofread in its entirety by a native speaker, with the aim of improving the editing of the language and style.
I attach the manuscript with the change control so that you can appreciate the change.
We hope that we have responded to your comments correctly.
Yours sincerely,
Irati Leon

Round 2
Reviewer 2 Report
The authors satisfactorily addressed most of the comments